# Stress Perception and Coping Strategies of Students on Both Sides of the EU’s Eastern Border during the COVID-19 Pandemic

**DOI:** 10.3390/ijerph191610275

**Published:** 2022-08-18

**Authors:** Andrei Shpakou, Elżbieta Krajewska-Kułak, Mateusz Cybulski, Izabela Seredocha, Anna Tałaj, Małgorzata Andryszczyk, Ewa Kleszczewska, Anna Szafranek, Beata Modzelewska, Ihar A. Naumau, Andrei Tarasov, Ludmila Perminova, Rafał Modzelewski

**Affiliations:** 1Department of Integrated Medical Care, Faculty of Health Sciences, Medical University of Bialystok, 15-089 Bialystok, Poland; 2Faculty of Health Sciences, Academy of Medical and Applied Social Sciences in Elblag, 82-300 Elblag, Poland; 3Department of Health Care, Prof. Edward F. Szczepanik State Vocational College in Suwałki, 16-402 Suwałki, Poland; 4Faculty of Social and Human Science, Lomza State University of Applied Sciences, 18-400 Lomza, Poland; 5Department of Biophysics, Medical University of Bialystok, 15-222 Bialystok, Poland; 6Department of General Hygiene and Ecology, Grodno State Medical University, 230009 Grodno, Belarus; 7Department of Pediatrics and Preventive Medicine, Medical Institute, Immanuel Kant Baltic Federal University, 236041 Kaliningrad, Russia; 8Department of Foreign Languages, Medical University of Bialystok, 15-222 Bialystok, Poland

**Keywords:** university students, pandemic COVID-19, anti-pandemic measures, stress, coping strategies

## Abstract

The aim of the study was to compare the perception of stress and the characteristic coping-strategies among students in the context of the different anti-pandemic measures taken in Belarus, Poland, and the Russian exclave of Kaliningrad. A cross-sectional online survey using standardized questionnaires (Perceived Stress Scale—PSS-10 and Brief-COPE—Mini-COPE inventory) was conducted among 3113 students of seven universities in three neighboring regions on both sides of the eastern border of the EU. The groups that are the most prone to stress are the Polish and Russians students. Among the students from Belarus, 122 (13.7%) have high levels of stress symptoms. Among the respondents from Poland—238 (19.4%), and 191 (19.2%) from Kaliningrad have high levels of stress, respectively. The different approaches of the authorities to the COVID-19 pandemic diversified the choice of students’ stress coping strategies. The behavior of the students from Kaliningrad and Poland was similar. The Belarusian students used active coping strategies less often, while an avoidance-focused style, and denial were more frequent. The neglect of restrictive anti-pandemic measures by the Belarusian students was manifested by a higher incidence of disease and minimal use of vaccinations.

## 1. Introduction

Infectious diseases, such as COVID-19, may have an immense influence on youth mental health [1]. In March 2020, the World Health Organization (WHO) declared the outbreak of COVID-19 as a pandemic [2]. It is a global health crisis that has impacted on daily life, due to the policies created to contain the outbreak [3]. Strict rules to protect the population from the virus, including social and physical distancing, were introduced as the countermeasures. However, the epidemiological situation varied from country to country, and the WHO recommendations were not universally accepted as guidelines, which also influenced the variation in coping strategies across countries [4].

In order to slow down the rapid spread across and within countries, many governments responded with strict measures, including lockdowns with school and work-place closures, self-isolation, social distancing, border closures, and restrictions in travel to reduce the transmission of the virus [5]. In response to the outbreak and spread of the pandemic, different countries and even individual regions within Europe followed different strategies.

Poland, as with most European countries, introduced severe quarantine measures (lockdown) in order to control the spread of the infection. Thus, the following measures were taken: person-to-person contacts were limited by introducing social distancing; unnecessary travel was banned; schools, universities, and sports facilities were temporarily closed; remote work was introduced.

In the Russian exclave of Kaliningrad and elsewhere in Russia, anti-pandemic measures were similar. However, there were several differences. Russia denied the existence of the virus for a long time, then downplayed its possible effects, and half-way and inconsistent steps were taken. The self-isolation regime was delayed, many organizations switched to remote work, mass events were banned, entertainment venues were closed, and state support (financial, social, medical) was implemented to maintain the well-being and health of Russian citizens [6].

In contrast to these countries, Belarus did not adopt a similar anti-pandemic strategy [6], denied the presence of the virus, and took no measures in the first few months of the pandemic. The citizens of Belarus were recommended to maintain the normal organization of their social lives, without serious restrictions on travel. No restrictions on social contact were introduced, while the population was persistently informed about the need to comply with the safety measures, which were to minimize panic, relieve anxiety, and reduce the stress and psychological burden on society [7]. Mandatory Non-Pharmacological Interventions (NPIs) remained the only forms of pandemic control in the country. Available NPIs included some contact tracing, 14-day self-isolation for laboratory-confirmed close contacts and for those returning from abroad, delayed school opening times, hybrid teaching at some universities, and an increased use of public transport to avoid congestion [8]. Since the social distancing measures were not strictly implemented or enforced, it was up to individuals to decide whether and how they would change their behavioral patterns.

The strong anti-pandemic measures, such as those taken in many parts of the world, on the one hand reduce the basic risk of the spread of the virus, but on the other hand, may not lead to the development of sufficient herd immunity, which may increase the risk of further pandemic waves [7]. The various strategies to counteract the spread of COVID-19 both in Poland and Russia (lockdown and its components) and in Belarus (no or minimal restrictions) can be considered to be the cause of an extremely stressful situation, accompanied by the lifestyle changes of numerous students. The significant changes in the way of life and work of students, as well as difficulties resulting from the introduced travel restrictions and social distancing orders, may possibly have a negative impact on their mental health [9].

According to Khozaeil and Carbon [10], even before the COVID-19 pandemic began, mental health was an extremely important and critical area of research. It is estimated that more than 10% of the world’s youth have mental disorders [11]; additionally, especially in a pandemic crisis it is important to focus on mental disorders and the further potential deterioration of mental health. Over the last few decades, research has been focusing less and less on the experience of stress, and more on the activity that a person undertakes to cope with stressful events, referred to as coping with stress. Under the conditions of an unstable state of society during a pandemic, a transformation and expansion of the repertoire of strategies for coping with stress was noted [12,13]. Stress coping strategies refer to the behavioral and cognitive efforts that help reduce the negative manifestation of a stressful situation that significantly reduces individual life resources [14]. Education about positive thinking, active coping, and social support could be beneficial for dealing with a decrease in mental health due to the COVID-19 pandemic. By choosing different coping strategies during various anti-pandemic measures, it is possible to consider the stressful situation caused by the pandemic not only as a negative component of the lifestyle, but also as a challenge and the possibility of active counteraction through the search for new ways to overcome it [15,16].

Students are a high-stress population who experienced deteriorated mental well-being during the pandemic. Some of the categories of students are at an increased risk of developing high stress [17,18,19]. Several studies reported younger age as a risk factor for mental health problems amid COVID-19 [20,21,22].

Studying stress coping in a comparative aspect, focusing on identical populations in closely situated cities in neighboring countries, where different methods of pandemic control were applied, shows great promise [23]. The students, as a rapidly adapting population, are an interesting target group for comparative research. The optimal way for such studies to be completed is by cross-border cooperation in the framework of joint virtual scientific teams (e-Science), using identical (unified) methods [24]. The set goals can be realized in the border region of Belarus (BY), Poland (PL), and Russia (Kaliningrad region) (RU), considering the region as an external eastern border of the European Union and the respondents as representatives of closely related cultures (having quite a lot in common in lifestyle and the cultural component). We intend to research the peculiarities of the way students perceive the situation in the context of various anti-epidemiological measures in a region where three countries, including the eastern border of the European Union, converge.

Purpose of the work: to study stress perception and the characteristic ways of overcoming stress, among university students on both sides of the EU’s eastern border in the context of the implementation of anti-pandemic strategies in Belarus, Poland, and the Russian exclave—Kaliningrad.

The research on the Belarusian population could contribute to a better understanding of the relation between the COVID-19 outbreak and alterations in mental health, since the measures taken against the pandemic in Belarus differ greatly from the ones applied in most countries. It is important to monitor the mental health of young people in the era of the ongoing COVID-19 pandemic; however, to date no such studies were conducted on the population of the students in Eastern Poland, Western Belarus, and Kaliningrad, and this is probably the first study of this kind among university students on both sides of the EU’s eastern border. In this study, we measured the perceived stress in response to the COVID-19 pandemic among Belarusian, Polish, and Russian students.

Promoting multi-disciplinary mental health research. COVID-19 brought unique challenges to all of the aspects of health and well-being, which include social, cultural, educational, occupational, economic, political, and other dimensions of human lives [25].

## 2. Materials and Methods

### 2.1. Study Design and Setting

This study is based on a cross-sectional survey carried out between January and February 2022, as a part of an international multi-center research project—The COVID-19 Coping Study of Students from East Europe (SEECoping-S). The sociological survey was conducted among the students from partner universities on both sides of the eastern border of the European Union (three countries). To collect information, an online survey form was developed in Russian, Belarusian, and Polish in electronic form using the Google Forms platform. An invitation to participate in an online survey was distributed through targeted advertising, including an e-learning platform (Moodle), Skype, Microsoft Teams, and university social networks. The proposed information resources were avail-able to students and were widely used in teaching during the COVID-19 pandemic.

All of the students attending a selected university from selected faculties in the three countries were eligible to be included in the research. The inclusion of a medical education center from each country was mandatory. An additional education center (non-medical-pedagogical) from each country was also randomly selected for inclusion. In West Belarus (BY) these are the universities (Grodno, Brest), East Poland (PL) (Elblag, Suwalki, Bialystok, Lomza), and Russia (RU) (Kaliningrad). The study used non-repeated random sampling, whereby a respondent only answered the questionnaire once. The beginning of the questionnaire included the information about the purpose and anonymity of the study, as well as the opportunity to withdraw from the study at any stage.

Permission was obtained from the leadership of the universities participating in the study to conduct an anonymous survey of the students. The research was conducted in accordance with the Declaration of Helsinki and its subsequent amendments. The participation was entirely voluntary, anonymous, and consensual; the informed consent from the participants was obtained prior to the survey being administered. No financial incentives were offered or provided for participation. The survey did not collect any identifiable information from the participants. The ethical permission to conduct the study was obtained from the Bioethical Review Board at the Medical University of Bialystok, Poland (document number: (APK.002.1932.2022).

An anonymous Internet survey was conducted among 3113 participants: the full-time students of two–five courses of seven universities (female *n* = 2318; male *n* = 795). The ratio of men to women in the samples surveyed was 1:3 to 1:4, reflecting the general trend in the ratio of men to women in medical and pedagogical faculties at universities in the three countries. The relationship by gender was maintained in all of the three represented countries. Their average age was from 18 to 33 years old ($\overset{-}{X}$ = 21.1; SD = 3.71) with a significant bias towards the younger participants. There were *n* = 889 participants in the Belarusian sample, *n* = 1228 in the Polish sample, and *n* = 996 in the Kaliningrad sample. No data were missing, since the online platform did not allow for the submission of incomplete questionnaires.

### 2.2. Study Questionnaire

The Perceived Stress Scale—PSS-10 (Russian and Polish versions) was used to determine the level of psychological stress and distress [26]. The participants were asked to rate their stress levels over the past month. The PSS-10 contains ten questions on a five-point scale from 0 to 4, and the scores of items 4, 5, 7, and 8 were reversed. The higher the score, the greater the feeling of stress (linear relation) [27].

Initially, the results of the PSS-10 were assessed using the following subscales: “Overload”, which measures the subjectively perceived level of tension in the situation; “Perceived stress”; and “Stress response”, which determines the level of effort made to overcome stress. The individual scores on the PSS can range from 0 to 40, with higher scores indicating higher perceived stress: scores ranging from 0–13 would be considered low stress; scores ranging from 14–26 would be considered moderate stress; and scores ranging from 27–40 would be considered high perceived stress. Cronbach’s (α) reliability analysis was used in order to verify the internal consistency of the questionnaire: the reliability of the tool was assessed as α = 0.856, which is a satisfactory level of reliability.

In order to assess the students’ stress coping strategies, the Mini-COPE questionnaire was used to examine the degree of coping with stress in a difficult situation. The Mini-COPE Questionnaire is the version of Brief-COPE [28]. The questionnaire was adapted into Polish by Juczynski and Oginska-Bulik [29], and into Russian by Rasskazova et al. [30]. The respondents, evaluating their choice, noted that it helps to cope with emotions, protect themselves and loved ones, reduce risks and threats, and prepare for the unknown and further developments.

When completing the questionnaire, a phrase was added to the general version of the instructions: “Please answer, how often do you use each of these options in a pandemic situation that has affected everyone in one way or another—to cope with your worries, protect yourself and your loved ones, reduce risks and threats”. The questionnaire began with the words: “When I am in a very difficult situation related to the impact of the COVID-19 pandemic, I usually…”, and included 28 statements describing 14 situation-specific coping strategies. There were two assertions for each strategy. Each statement is graded on a four-point Likert scale: 0—never do this; 1—rarely; 2—often; 3—almost always. Each of the 14 coping strategies is measured by two items on the average number of points. The higher the score, the more often the individual uses this strategy to resolve a stressful situation. The 14 strategies are: Scale 1—active coping (actions to eliminate, reduce the stressor or its consequences); Scale 2. Planning (thinking about and planning what to do); Scale 3. Positive reframing (thinking about a negative or challenging situation in a more positive way); Scale 4. Acceptance (accepting the situation as irreversible, which you need to get used to); Scale 5. Humor (as a way to soften unwanted emotions); Scale 6. Religion (as a source of emotional support and a pointer to a positive reappraisal); Scale 7. Use of emotional support (sympathy, understanding, moral support); Scale 8. Use of instrumental support (the desire to get advice, help, or reliable information); Scale 9. Self-distraction (avoiding thoughts about the situation by engaging in other activities); Scale 10. Denial (denial of the reality of a stressful situation, ignoring it); Scale 11. Venting (focus on emotions and their manifestation, worry about your emotions, a tendency to discharge them); Scale 12. Substance use (use of alcohol or other psychoactive drugs); Scale 13. Behavioral disengagement (“helplessness”, “submission”, and “refusal of efforts”); and Scale 14. Self-blame. The α-Cronbach coefficient amounted to α = 0.893 and yielded satisfactory results.

All of the 14 coping strategies were grouped into three integral ones. Thus, the active strategy included active coping, planning, and positive reframing (Problem-Focused Coping—scales 1–3). The Emotion-Focused Coping support-seeking strategy included religion, the use of emotional and instrumental support, the inability to contain emotions, venting them, and a strategy of self-condemnation (scales 6–8, 11, 14). The Avoidant Coping is the scenario that consisted of acceptance or denial of the situation, self-distraction with other activities, behavioral withdrawal, and a tendency to use alcohol and psychoactive substances (scales 4, 5, 9, 10, 12, 13) [28].

### 2.3. Statistical Analysis

The statistical software package Statistica 13 PL (StatSoft, Tulsa, OK, USA) was used for the statistical processing and for the search for significant dependencies. The quantitative variables had a distribution that differed from the norm (an assessment of the correspondence of the obtained values to the normal distribution of the variation series using the Shapiro–Wilk *W*-test).

A pairwise comparative analysis between the groups in the three countries was carried out using the independent sample *t*-test. This test assumes that the samples come from populations in which the variable under analysis follows a normal distribution. However, moderate deviations from normality are not problematic, especially in large samples—which is the case in this study. Therefore, in the processing and interpreting the results, the standard indicators were used: the arithmetic mean ($\overset{-}{X}$), and the standard deviation (±SD). To compare the coping strategies between the two groups, the Mann–Whitney U test was additionally used. The Kruskal–Wallis test was used to compare the three groups. For the qualitative variables, the frequencies and percentages were used. Pearson’s chi-square χ^2^ tests were used. The interval estimation of the statistical parameters was determined using 95% confidence intervals. For all of the analyses, the *p* values < 0.05 were considered statistically significant.

## 3. Results

### 3.1. Characteristics of the Sample

The participants from Belarus and Russia were younger than the Polish ones, $\overset{-}{X}$_BY_ = 18.8; SD = 1.70 and $\overset{-}{X}$_RU_ = 19.6; SD = 1.86 vs. $\overset{-}{X}$_PL_ = 23.9; SD = 4.04, the Kruskal–Wallis *H* test: *H* = 1353.6, *p* = 0.0001. This is due to the fact that in Poland, people receive higher education at a later age. The data were divided into three groups, based on the number of students from each region who took part in the study. The results of the two-way ANOVA for the respondents according to country, faculty (medical and non-medical), gender (male, female), age, and coronavirus-related factor is demonstrated in Table 1.

A total of 24.8% of the study (N = 771) participants noted that they had had COVID-19 (approximately the same number of men and women). The indicators included in the study were consistent with a recognition of the fact that a diagnostic confirmation of COVID-19 had taken place. The statistical differences by country are significant. Among the Belarusian students, the number of vaccinated participants was the lowest of all three groups.

### 3.2. Perceived Stress Scale (PSS-10)

A total of 17.7% of the students experienced stress (PSS-10 greater than 27). The minimum level was recorded among Belarusian students (13.7% vs. 19.4% and 19.2%, respectively, among Polish and Russian students). The results of the assessment in the “Overload” subscale confirmed a lower level of stress among students from Belarus (10.9 ± 5.75 vs. 12.6 ± 5.50 and 12.1 ± 6.23, respectively). The average levels of perceived stress of the entire study group, grouped into “Overload”, “Perceived stress”, and “Stress response” categories, are presented in Table 2.

The majority of the students in the three countries reported moderate perceived stress. Overall, 11.1%, 71.2%, and 17.7% of the respondents reported low, moderate, and high perceived stress. If the students from Poland and Kaliningrad have a similar distribution of low, medium, and high levels of stress, the Belarusian students differ in that they twice as frequently have low levels of stress. This is confirmed by the higher levels of stress among the Polish and Russian students.

The subscale “Stress response” did not adequately demonstrate the differences in the use of coping strategies by region, education profile, and gender. By examining the coping strategies in more detail, we were able to examine these differences. The women were more likely than the men to be characterized as experiencing high levels of stress. Consequently, higher levels of stress were associated with being a woman and living in a country or area with stricter COVID-19 anti-pandemic measures.

### 3.3. The Choice of Coping Strategies

A comparative analysis was conducted of the stressful situation management by the students in Belarus, Poland, and Kaliningrad. The statistically significant differences occurred for all of the variables, except for instrumental support. The respondents indicated that they most often fight stress by active coping (2.04 ± 0.72), i.e., taking action to improve the situation; planning (1.94 ± 0.74), i.e., thinking about and planning what to do; acceptance (1.69 ± 0.74), i.e., accepting the situation and learning how to live with it; positive reframing (1.69 ± 0.81), i.e., thinking about a negative or challenging situation in a more positive way, and seeking instrumental support (1.74 ± 0.81), i.e., seeking and receiving advice and help from others.

Table 3 shows the distribution of the frequency of the choice of coping strategies among the students surveyed and the significant differences in the three countries.

Overall, the Russians and Polish respondents took more action in the stressful situations. Avoidance, blaming oneself, denial of the reality of a stressful situation, ignoring it, turning to religion and behavioral withdrawal (“helplessness”, “submission”, and “refusal of efforts”) were among the rarely used strategies.

A series of one-way ANOVA was performed separately for the three coping styles: activity-oriented; emotion-oriented; and avoidance-oriented. When choosing the three integral coping strategies, it was noted that the students almost equally and more often resorted to active coping (“Active”) (Figure 1).

The coping style that focused on an active increase in the resistance to stress consists of problem-solving by changing the stressful situation, through prioritizing active measures. The Belarussian students had significantly higher levels of avoidance strategies. The least frequent strategies in the three countries were: substance use; and religious coping. The students from Kaliningrad very rarely used religion as a coping strategy. At the same time, the students from Poland and Belarus more often cited the use of religion as a coping strategy. The emotional support strategy had a protective effect against psychological stress symptoms, but it was the only one to show gender differences. The female students were characterized by a higher use of emotional support strategies. The medical students were more likely to use active coping, planning, and use of emotional support strategies than the non-medical students.

## 4. Discussion

The impact of the COVID-19 pandemic on the mental health of the population has been scientifically analyzed as a multifactorial process. [31]. The effectiveness of adaptation to the new conditions resulting from the psycho-traumatic consequences of the COVID-19 pandemic is determined by the influence of both external circumstances and personal traits [32,33]. Detailed and interdisciplinary research is necessary in order to conduct a deep analysis of the situation [34]. When interpreting the results of the impact of the COVID-19 pandemic, the country-specific methods of countering the pandemic should be taken into account [5]. The data collected from the various universities across the world suggest that university students experience higher levels of stress than the general population [35].

In this study, we have shown the prevalence of perceived stress and choice of coping strategies in the neighboring regions of the three countries and their changes in a total sample of 3113 students.

The regions (Western Belarus, Eastern Poland, and the Kaliningrad Region of Russia) are geographically close. They were notable for the severity of the measures taken against the COVID-19 pandemic. The study was conducted with high methodological standards, including representative samples, which allowed them to be generalized to the young adult population.

The participants in this study saw themselves in a stressful pandemic situation and under a genuine threat to their mental health. We found differences in the standardized indicators of perceived stress and adaptation depending on the severity of the anti-pandemic action announced by the state, and we took into account the important factors influencing this phenomenon. On the one hand, Belarus -adopted an alternative approach to fighting the pandemic compared to most countries. As a result, the groups of people who were not directly exposed to COVID-19 were exposed to less stress. On the other hand, some of the groups of people (taking into account gender and future occupations) [36,37] who were exposed to the pandemic may have perceived government action as insufficient, which increased the stress.

As a result, the students from Poland and Kaliningrad experienced higher levels of stress. Almost 20% of the survey respondents reported symptoms of severe stress. In a similar comparative study conducted in Germany and Great Britain, 25% of the respondents reported a subjective severity of general psychopathological symptoms [38]. Among the students from Belarus, this indicator was twice as low.

The people’s mental health assessment scores usually relate to their ability to cope with stressful situations. Coping can be understood as a process that is key in managing stress. In this process, the individual makes a cognitive and behavioral effort to manage both the external and internal sources of stress [5]. Liang et al. [1] assessed the magnitude and predictors of mental health problems among adolescents and found that negative coping styles can lead to adverse effects on human health.

The relationship between stress and coping in university students is also interesting. Typically, students differ in their choice of using specific coping strategies, which does not mean that a person’s coping strategies are the same in all of the situations [39,40]. The differences in the choice of active and other coping strategies can be explained by anti-pandemic activities that limit normal activity in the activities of daily living limit or block movement due to social distance [41].

Our study demonstrates that the feelings of insecurity and sense of crisis [42] during a pandemic suggests that the students mainly choose problem-oriented coping strategies. These activities consist of planning and reformulating one’s own life situation. The second group of selected coping strategies confirms the theory formulated by Lazarus and Folkman [43], and is oriented towards emotions and acceptance. Denial, religious coping, and especially the use of psychoactive substances were the least frequently used by student youth to cope with stress during the pandemic, which is both consistent with [44,45] and contradictory to [46,47], and the results of other studies [48,49]. A less frequent use of alcohol may be a result of social righteousness in the responses; however, it may also result from a significant reduction in the use of alcoholic beverages.

Comparatively, the students from Poland and Kaliningrad obtained higher scores in active coping, planning, seeking instrumental support, and significantly less often re-sorted to denial and ignoring the problem and behavioral withdrawal (“helplessness”, “submission”, “abandonment of effort”). We found the most interesting results for the humor and religion strategies. Humor was the only exception among all of the avoidance and support-seeking/emotion-oriented strategies, which was significantly more frequently used by the Russian students. However, the greatest difference was present in the reliance on religion among all of the strategies, favoring the Polish and Belarusian students.

The coping profile of the students from Belarus in our sample was characterized by a focus on their readiness to cope with a critical, extreme situation, but with less pronounced manifestations of active coping strategies. Generally speaking, the students used problem-oriented coping strategies more often, but problem avoidance prevailed among the Belarusian students. It may be related to their insufficient access to information about COVID-19 and its consequences. This manifested itself in minimal denial of the situation, and in a reluctance to focus on the behavioral and mental shortcomings. Accordingly, the pandemic and the severe measures to combat it have contributed to the acceptance of the real phenomena and an appreciation of the importance of the problem [50].

The stricter the restrictions, the more often the participants of the study chose active coping strategies. The demonstrated tendency to choose coping strategies by students may be characterized by the characteristics of students, and their readiness to work in critical and extreme situations. Similarly to pre-pandemic research, our study confirmed that women experienced higher levels of stress [51]. A detailed analysis revealed differences between the sexes in the application of some of the specific coping strategies, which was also confirmed by other authors [49]. In this study, the female respondents focused on active emotional expression, social and instrumental support, positive reformulation and personal growth, acceptance, and mental avoidance of the problem to a greater extent compared to the males. In contrast, the males tended to choose active handling strategies, rather than attempting to rethink a stressful situation in a positive manner. One study [51] demonstrated the ineffectiveness of mental care and emotion-oriented coping strategies in a pandemic situation, which may lead to increased psychological stress among the university students. However, without longitudinal studies, it is difficult to assess the inconsistency in the application of coping strategies in a pandemic situation, which, however, does not exclude the need to develop preventive measures to maintain mental health and well-being, taking gender into account. The differences in the frequency of using coping strategies by students of the same sex from different university centers also make it possible to determine the influence of the factor of differentiation of preventive action. The Belarusian students more often reported a lack of conviction in the reality of the situation, which was accompanied by a denial of reality, which may be related to the way that the information about the epidemiological situation was presented to the public, and a less severe change in lifestyle. The differences were observed in the treatment of religion and faith, which reflects the socio-cultural characteristics of the respondents (in Western Belarus and in Poland, the traditional Catholic faith is strong). Furthermore, the male respondents demonstrated a low level of concentration on emotions, and a frequent use of active coping strategies and humor. The research shows not only the similarities in the targeting behaviors related to coping with the pandemic (emotional reactions and distraction from negative situations in women and finding new opportunities and looking for positives in men), but also the differences in the strategies for their implementation, due to the assessment of the situation and one’s own opportunities [52].

It is worth noting here, however, that it is not always possible to directly compare the results of our study with those obtained by other researchers globally, because the ways of coping with stress have not been systematized so far, and therefore there is no universally used classification of coping strategies. The goal of our study was to document the students’ coping strategies, while focusing on the possible impact of internal and external factors in the event of the COVID-19 pandemic. The dominant types of behavior and the consequences of the identified differences, as well as the evaluation of the functionality of the applied coping strategies require further interdisciplinary research. The anti-pandemic measures, such as the ones undertaken by many countries of the world, decrease the underlying risk of the spread of the virus, but do not result in the development of a sufficient collective immunity, which poses a risk of subsequent pandemic waves. Additional research is needed on the impact of the late stages of the pandemic on the mental health of students, as the effects of this difficult situation may persist long after the most intense period of the pandemic has passed.

### Strengths and Limitations

The strengths of the current study include an examination of stress, strategies of coping with stress in a large sample of students during the pandemic, taking into account various anti-epidemic preventive measure. The essence of such a comparison was that in neighboring regions, the authorities applied different approach to the COVID-19 pandemic—full lockdown in Poland and Kaliningrad, and no or minimal restrictions imposed by the government in Belarus. To our knowledge, such a comparison has not yet been carried out by other researchers, and it is probably the first study of this kind among university students on both sides of the EU’s eastern border. The current results might also serve as a point of reference and comparison for further studies on coping strategies among students in other regions. The main limitations of the study concern the data collection methods. The data collection was carried out at a certain point in time, and not longitudinally. A limitation of the current study is the high proportion of young women, which could potentially result in a bias. However, this proportion of men and women is typical of the medical and pedagogical faculties in the three countries. We used a self-reporting survey without clinical in-person verification. However, by using a completely voluntary and anonymous format, as well as standardized questionnaires, we minimized the potential effects. It allows us to form hypotheses for future research.

## 5. Conclusions

The impact of COVID-19 on the mental health of the Belarusian students is not comparable to the impact in the neighboring regions in Poland and the Kaliningrad region of Russia. The differences in the stress prevalence and coping strategies were found between the countries, and these differences are probably related to the different approach of the authorities to the COVID-19 pandemic. During the COVID-19 pandemic, the students who were surveyed dealt with stress mainly through active coping, planning, acceptance, positive reassessment, and seeking emotional support. The groups most exposed to stress were the Polish and Russian students. The effect of the minimal anti-pandemic restrictions in Belarus also increased the risk of stress among the Belarusian students. The neglect of restrictive anti-pandemic measures by the Belarusian students was manifested in a higher morbidity and minimal use of vaccination. It was also shown that the Belarusian students more often used coping strategies based on helplessness. They used active coping strategies less often, and more often turned to the avoidance-oriented style, behavioral detachment, and denial. The Polish students, in addition to active coping, turned to religion more often, but used humor and alcohol less frequently. The behavior of the students from Kaliningrad was similar to the Polish students’ methods of coping with stress. The support-seeking/emotion-oriented strategies were used with equal frequency in the three groups. These results are encouraging and point to the normative adaptation processes that young adults go through when faced with significant stress. Going forward, these results may help determine the relevance of stress and mental health monitoring in vulnerable groups, in order to develop specific prevention and intervention programs to improve stress coping skills.

## Figures and Tables

**Figure 1 ijerph-19-10275-f001:**
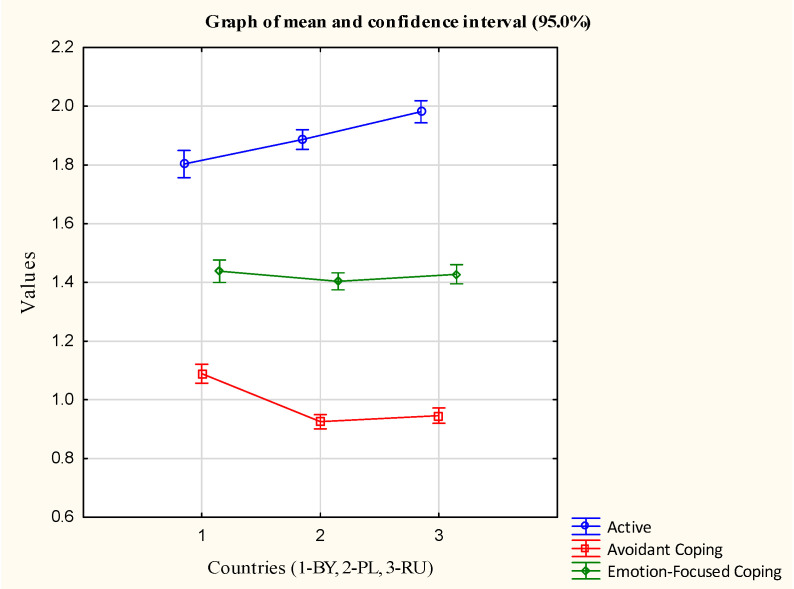
Scores for coping with stress (Mini-COPE) in three countries.

**Table 1 ijerph-19-10275-t001:** Characteristics of study participants.

	Universities in the Countries			Total Sample N = 3113 (100%)	
	Belarus, N = 889 (28.6%) (BY)	Poland, N = 1228 (39.4%) (PL)	Kaliningrad Region of Russia, N = 996 (32.0%) (RU)		
Male, *n*; %, (95% CI)	204; 22.9 (20.2–25.7)	281; 22.9 (20.5–25.2)	310; 31.1 (28.2–34.0)	795; 25.5 (24.0–27.1)	
Female, *n*; %, (95% CI)	685; 77.1 (74.3–79.8)	947; 77.1 (74.8–79.5)	686; 68.9 (66.0–71.8)	2318; 74.5 (72.9–76.0)	
Age, mean, years ± SD		18.8 ± 1.70	23.9 ± 4.04 *	19.6 ± 1.86	21.1 ± 3.71
Medical, *n* (%)		513 (57.7)	722 (58.8)	541 (54.3)	1776 (57.1)
Non-medical, *n* (%)		376 (42.3)	506 (42.2)	455 (45.7)	1337 (42.9)
Vaccinated against COVID-19, *n*; % (95% CI)		373; 42.0 (38.7–45.2)	757; 61.6 (58.9–64.4)	831; 83.4 (81.1–85.7)	1961; 63.0 (61.3–64.7)
	*** P*_BY-PL_ < 0.01; *P*_BY-RU_ < 0.01; *P*_PL-RU_ < 0.01; *P*_BY-PL-RU_ < 0.01 (K-Wt)				
Diagnosed with COVID-19 (infection with SARS-CoV-2), *n*, % (95% CI)	273; 30.9 (27.7–33.7)	244; 19.9 (17.6–22.1)	254; 25.5 (22.8–28.2)	771; 24.8 (23.3–26.3)	
	*** P*_BY-PL_ < 0.01; *P*_BY-RU_ < 0.01; *P*_PL-RU_ < 0.01; *P*_BY-PL-RU_ < 0.01 (K-Wt)				

Note: N is the number of observations, % is the percentage of the total number of study participants in a given group 95% CI—95-percent confidence interval; SD—standard deviation; K-Wt—value of the Kruskal–Wallis test; *—differences in age between Polish students and students from Belarus and Kaliningrad are significant (*p* < 0.05); **—differences in vaccinated against COVID-19 and diagnosed with COVID-19 between each country are significant (*p* < 0.05).

**Table 2 ijerph-19-10275-t002:** Summary of data describing differences in scores obtained by students from three regions on the Perceived Stress Scale and its subscales (Mean score).

Subscale and PSS Score		Universities in the Countries			Total Sample N = 3113 (100%)
		Belarus (BY)	Poland (PL)	Kaliningrad Region of Russia (RU)	
Overload subscale		10.9 ± 5.75	12.6 ± 5.50	12.1 ± 6.23	12.0 ± 5.85
		K-Wt: *P*_BY-PL_ < 0.01; *P*_BY-RU_ < 0.01; *P*_PL-RU_ > 0.1			
Perceived stress		19.5 ± 7.86	21.6 ± 6.20	21.4 ± 6.56	20.9 ± 6.88
		K-Wt: *P*_BY-PL_ < 0.01; *P*_BY-RU_ < 0.01; *P*_PL-RU_ > 0.1			
Stress response		8.6 ± 3.80	9.0 ± 3.18	9.3 ± 4.0	9.0 ± 3.51
		K-Wt: *P*_BY-PL_ > 0.1; *P*_BY-RU_ < 0.01; *P*_PL-RU_ > 0.1			
PSS Score, (*n*; %, 95% CI)	Low stress	158;17.8 (15.3–20.3)	101; 8.2 (6.7–9.8)	87; 8.7 (7.0–10.5)	346; 11.1 (10.0–12.2)
	Moderate stress	609; 68.5 (65.5–71.6)	889; 72.4 (69.9–74.9)	718; 72.1 (69.3–74.9)	2216; 71.2 (69.6–72.8)
	High perceived stress	122; 13.7 (11.5–16.0)	238; 19.4 (17.2–21.6)	191; 19.2 (16.7–21.6)	551; 17.7 (16.4–19.0)
		χ^2^_BY__-PL_ = 49.2; *p* < 0.01; χ^2^_BY__-RU_ = 38.8; *p* < 0.01; χ^2^_PL__-RU_ = 0.2; *p* > 0.1			

Note: K-Wt—value of the Kruskal–Wallis test.

**Table 3 ijerph-19-10275-t003:** Descriptive statistics for measures of coping strategies among students ($\overset{-}{X}$ ± SD).

Coping-Strategy (Scale Number)	Belarus (BY)	Poland (PL)	Kaliningrad Region of Russia (RU)	Total Sample	Comparison, *p **			*p ***
					BY-PL	BY-RU	PL-RU	
1. Active Coping	1.89 ± 0.76	2.06 ± 0.70	2.14 ± 0.70	2.04 ± 0.72	0.001	0.001	0.01	0.001
2. Planning	1.81 ± 0.79	1.95 ± 0.69	2.04 ± 0.73	1.94 ± 0.74	0.001	0.001	0.01	0.001
3. Positive reframing	1.71 ± 0.83	1.65 ± 0.75	1.76 ± 0.85	1.69 ± 0.81	N/S	N/S	0.01	0.01
4. Acceptance	1.60 ± 0.79	1.73 ± 0.69	1.72 ± 0.75	1.69 ± 0.74	0.001	0.001	N/S	0.001
5. Humor	1.58 ± 0.73	1.03 ± 0.77	1.71 ± 0.94	1.40 ± 0.88	0.001	0.001	0.001	0.001
6. Religion	0.90 ± 0.91	0.96 ± 0.95	0.59 ± 0.86	0.82 ± 0.92	N/S	0.001	0.001	0.001
7. Use of emotional support	1.83 ± 0.85	1.80 ± 0.79	1.77 ± 0.84	1.87 ± 0.83	N/S	0.001	0.001	0.001
8. Use of instrumental support	1.69 ± 0.82	1.75 ± 0.78	1.8 ± 0.84	1.74 ± 0.81	N/S	N/S	N/S	N/S
9. Self-distraction	1.07 ± 0.66	0.96 ± 0.62	0.99 ± 0.62	1.0 ± 0.63	0.001	0.01	N/S	0.001
10. Denial	1.03 ± 0.78	0.75 ± 0.73	0.65 ± 0.73	0.80 ± 0.76	0.001	0.001	0.001	0.001
11. Venting	1.44 ± 0.70	1.31 ± 0.72	1.51 ± 0.72	1.41 ± 0.72	0.001	0.05	0.001	0.001
12. Substance use	0.58 ± 0.78	0.38 ± 0.67	0.35 ± 0.65	0.40 ± 0.70	0.001	0.001	N/S	0.001
13. Behavioral disengagement	0.87 ± 0.70	0.67 ± 0.66	0.61 ± 0.64	0.71 ± 0.67	0.001	0.001	N/S	0.001
14. Self-blame	1.33 ± 0.85	1.21 ± 0.85	1.28 ± 0.89	1.26 ± 0.87	0.001	N/S	N/S	0.001

Note: *p* *—test probability value calculated using *t*-test. **—test probability value calculated using Kruskal–Wallis *H* test. N/S—differences are not significant (*p* > 0.05).

## Data Availability

The data that support the findings of this study are available on request from the corresponding author. The data are not publicly available due to privacy restrictions.

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
