# Peer review of "Stress Perception and Coping Strategies of Students on Both Sides of the EU’s Eastern Border during the COVID-19 Pandemic"

_ijerph, 2022, doi:10.3390/ijerph191610275_

Round 1

Reviewer 1 Report

The work has improved sufficiently and I think it can be published with a good quality.

Author Response

Dear Reviewer,

Thank you very much for your valuable comments on our paper titled “Stress Perception and Coping Strategies of Students on Both Sides of the EU’s Eastern Border During the COVID-19 Pandemic”.

Thank you for your valuable comments and suggestions.

This article is the result of a great international project. We are very pleased that the results will be published in such a prestigious journal in which you serve as a reviewer.

Kind regards,

Andrei Shpakou, PhD, MD

Department of Integrated Medical Care, Faculty of Health Sciences, Medical University of Bialystok, Poland

Reviewer 2 Report

Dear Editor,
I really appreciate the opportunity to review the manuscript ijerph-1855053 entitled:
"Stress Perception and Coping Strategies of Students on Both Sides of the EU’s Eastern Border During the COVID-19 Pandemic"

The paper is very interesting and well-written, methodologically unexceptionable, and the new implementations provide a valid contribution to the work. Every requested correction has been done, and the manuscript is now suitable for publication

Author Response

Dear Reviewer,

Thank you very much for your valuable comments on our paper titled “Stress Perception and Coping Strategies of Students on Both Sides of the EU’s Eastern Border During the COVID-19 Pandemic”. We have taken into account all the comments and recommendations of the esteemed Reviewers and have made the necessary changes in the text of the article.

Thank you for your valuable comments and suggestions.

This article is the result of a great international project. We are very pleased that the results will be published in such a prestigious journal in which you serve as a reviewer.

Kind regards,

Andrei Shpakou, PhD, MD

Department of Integrated Medical Care, Faculty of Health Sciences, Medical University of Bialystok, Poland

This manuscript is a resubmission of an earlier submission. The following is a list of the peer review reports and author responses from that submission.

Round 1

Reviewer 1 Report

This manuscript presents the results of a cross-sectional study conducted during COVID-19 spreading in three different countries. The novelty of the study is mainly due to comparing the mental health status of three populations who share similar cultural origins and living places, but who were in fact subjected to different containment measures due to the pandemic. However, the study is really poor on both the methodological and analytical aspects.

In fact, the methods were poorly described, and they only consists in presenting two scales to the three student's samples. 

The statistical analysis is to me no-sense. The authors used arbitrary cutoffs to cut samples in stressed and non-stressed individuals and then conducted a series of a non-parametric tests comparing males vs females, students from different countries, and medical and non-medical students. All information is sparsed and confused, and most of the tables are not readable at all. For example, what is compared in Table 5 is completely obscure. Moreover, it seems like all the comparisons were significant, even when the reported means and SDs were comparable. This sounds strange.

The figures then are not informative in any way, except Figure 4. For example, what is plotted in Figure 1 is not comprehensible at all.

Starting from this point and from the obscureness of results, the discussion is mostly speculative and non informative also.

I would like to suggest to authors to completely re-write this paper, focusing on the differences between the three samples. Then, the differences between males and females or between different classes of students should be marginalized in the economy of the paper. Moreover, please use the linear score and conduct normal, correlative, and comparative statistics between variables. There is no need to use cutoffs and divide the sample, nor any real reasons to do this. Thus, please go for comparisons in stress between the three samples, then compare them for the coping strategies (I suggest using only the three coping styles for a higher clarity), and then assess if using different strategies leads to different stress levels, maybe overall samples. Then, comparing stress between sex and students' status could be done in parallel or in specific analyses. 

Reviewer 2 Report

Line 162: en el diseño del estudio dicen que se desarrolló un formulario de encuesta en línea, pero cómo consiguieron la autorización para enviar a todos los estudiantes de forma aleatoria el cuestionario, pues necesitan sabe su correo electrónico (deberían aclarar este procedimiento). ¿Y metieron todos los cuestionarios que posteriormente dicen que han utilizado (imagino que ya validados) en ese cuestionario?

Line 201: el Alfa de Cronbach para los tres cuestionarios, si el autor dice que pasaron tres cuestionarios diferentes.

In general:

- Was using so many and rather long questionnaires a limitation in answering all the responses?

- The study describes the use of the BRIEF COPE in the methodology, however, it is not included in the results or is not sufficiently visible to the reader.

Reviewer 3 Report

Dear Editor,
I really appreciate the opportunity to review the manuscript ijerph-1776039 entitled:
"Stress Perception and Coping Strategies of Medical and Non-Medical Students on Both Sides of the Eu's Eastern Border During the Covid-19 Pandemic."

I very much enjoyed reading the proposed study. The idea is undoubtedly good and the assessment of coping strategies in three different settings is interesting. The study was conducted with high methodological standards, including representative samples, unfortunately, although the data were collected well, and in an absolutely commendable amount, the exposition in the article is definitely confusing and needs to be totally revised.

In some sections the article seems to be a blanket endorsement of the actions of the Belarusian state, which is certainly not the place of a scientific journal, nor of the authors, to evaluate. I beg the authors to remove any sentence to this effect such as "Belarus has adopted an exceptional approach to fighting the pandemic than most countries." and to use a more technical tone to emphasize the data.

It is further stated that "In Belarus, state-imposed measures against the pandemic were exceptional in comparison to most of the neighboring countries of the European Union and Russia. Therefore, it was expected that in Belarusian students, the increased severity of the effects of the COVID-19 pandemic should not have a significant impact on mental health and the level 395 of stress would be minimized."; this affirmation if it can certainly be acceptable from a clinical and neurobiological point of view (less covid = less covid damage) does not take into account a large amount of scientific literature on the effects on the stress of any form of lockdown, which was exceptional and necessary to avoid covid deaths but left as relics so many cases of stress. I beg the authors to clarify this point.

Finally, in the Conclusions section, something profoundly different is apparently said including "The impact of COVID-19 on mental health in Belarusian medical and non-medical students are comparable to the impact in neighboring regions in Poland and the Kaliningrad region of Russia"

and

"The effect of minimal anti-pandemic restrictions in Belarus only slightly reduced the risk of stress in Belarusian students."

So the argument does not come across well. I suggest that the authors revise the writing of the article, which is absolutely valid and worthy of seeing these data published, but reported in a more concise and straightforward manner.